# Gross Total Resection Promotes Subsequent Recovery and Further Enhancement of Impaired Natural Killer Cell Activity in Glioblastoma Patients

**DOI:** 10.3390/brainsci12091144

**Published:** 2022-08-27

**Authors:** Cheng-Chi Lee, Jeng-Fu You, Yu-Chi Wang, Shao-Wei Lan, Kuo-Chen Wei, Ko-Ting Chen, Yin-Cheng Huang, Tai-Wei Erich Wu, Abel Po-Hao Huang

**Affiliations:** 1Department of Neurosurgery, Chang Gung Memorial Hospital, Linkou, Taoyuan City 33305, Taiwan; 2College of Medicine, Chang Gung University, Taoyuan City 33302, Taiwan; 3Department of Colon and Rectal Surgery, Chang Gung Memorial Hospital, Linkou, Taoyuan City 33305, Taiwan; 4Department of Neurosurgery, New Taipei Municipal TuCheng Hospital, New Taipei City 236027, Taiwan; 5Institute of Polymer Science and Engineering, National Taiwan University, Taipei City 10663, Taiwan; 6Department of Surgery, College of Medicine, National Taiwan University Hospital, Taipei City 100229, Taiwan

**Keywords:** glioblastoma, natural killer cells, immune function, interferon, gross total resection

## Abstract

**Highlights:**

Natural killer cell activity is dramatically impaired in patients with glioblastoma.Surgical resection of glioblastoma promotes redistribution of NK cell subsets and increases NK cell activity 30 days after surgery.Gross total resection rather than subtotal resection significantly recovers and further increases the impaired NK cell activity in patients with glioblastoma.

**Abstract:**

Glioblastoma is the most common primary malignant brain tumor, and median survival is relatively short despite aggressive standard treatment. Natural killer (NK) cell dysfunction is strongly associated with tumor recurrence and metastasis but is unclear in glioblastoma. NK activity (NKA) represents NK cell-secreted interferon-γ (IFN-γ), which modulates immunity and inhibits cancer progression. This study aimed to analyze NKA in glioblastoma patients to obtain a clearer overview of immunity surveillance. From 2020 to 2021, a total of 20 patients and six healthy controls were recruited. Peripheral blood samples were collected preoperatively and on postoperative days (POD) 3 and 30. Then, NKA was measured using the NK VUE kit. Although NKA decreased on POD3, it recovered and further significantly enhanced on POD30, with a nearly five-fold increase compared to baseline (*p* = 0.004). Furthermore, the percentage of CD56^bright^CD16^−^ NK cells decreased significantly on POD3 (*p* = 0.022) and further recovered on PO30. Subgroup analysis of extent surgical resection further revealed that the recovery of impaired NKA was attributable to gross total resection (GTR) rather than subtotal resection (STR). In conclusion, NKA is significantly impaired in glioblastoma, and GTR has demonstrated superior benefit in improving the suppressed NKA and increased CD56^bright^CD16^−^ NK subset in glioblastoma patients, which may be associated with subsequent patients’ prognosis. Therefore, the goal of performing GTR for glioblastoma should be achieved when possible since it appears to increase NKA cell immunity.

## 1. Introduction

Natural killer (NK) cells are granular lymphocytes of the innate immune system, and possess both innate and adaptive immune features [1,2]. Two main biological functions of NK cells for eliminating stressed, virus-infected, or malignant cells are direct NK cell-mediated cytotoxicity and indirect secretion of cytokines, such as interferon gamma (IFN-γ) [3,4,5]. In humans, NK cells are defined as CD3^−^CD56^+^ and/or CD3^−^CD16^+^ cells. In addition, NK cells can be further classified into two main subsets based on the expression level of CD16 and the cell surface density of CD56: the CD56^bright^CD16^−^ and CD56^dim^CD16^−^ NK cell subsets [6,7]. CD56^dim^CD16^+^ NK cells are the major circulating NK cell subset, comprising at least 90% of all peripheral blood NK cells, whereas approximately 10% of NK cells are CD56^bright^CD16^−^ subset [8]. CD56^bright^CD16^−^ NK cells have the capacity to secrete high amounts of cytokines such as IFN-γ but have less cytolytic activity. In contrast, CD56^dim^CD16^+^ NK cells possess significantly higher cytotoxic activity by producing much more perforin, granzymes and cytolytic granules. Although NK cell subsets can be determined based on their surface molecule repertoire under normal and different pathological conditions [9,10,11], the distribution of NK cell subsets in glioblastoma patients is still unclear.

Glioblastoma is the most common primary malignant brain tumor, accounting for 14.3% of all tumors and 49.1% of malignant central nervous system tumors [12,13]. The current standard of care for patients with glioblastoma is surgery, followed by a combination of radiation and chemotherapy. However, this aggressive therapeutic strategy has achieved only limited success, with a median overall survival (OS) of approximately 15 months and a median progression-free survival (PFS) of only approximately 6.2–7.5 months [14,15]. The dysfunction of immune cells, such as T cells and NK cells, has been recorded in association with physical trauma, such as thermal injury and surgery [16,17,18,19]. Moreover, evidence suggests that the immune dysfunction after surgery may be implicated in disease recurrence, metastasis and death [20]. It is still unknown, however, whether cranial surgery for glioblastoma affects the distribution of NK cell subtypes and their activity.

Many studies have demonstrated that NK cells are regarded as the major IFN-γ producer among peripheral blood mononuclear cells (PBMCs) for innate and adaptive immune responses [21,22]. NK cell-secreted IFN-γ is not only associated with cancer cell growth, apoptosis and tumor suppression, but is also correlates strongly with NK cell cytotoxicity [23,24,25]. In recent years, NK cell activity (NKA) has been widely measured by detecting secreted IFN-γ from NK cells in the ex vivo stimulated PBMC [26,27,28]. Therefore, the aim of this study was to investigate the influences of cranial surgical resection of glioblastoma on NKA and the distribution of NK cell subsets. Given the superior benefit of gross total resection (GTR) compared to subtotal resection (STR) in improving the survival outcomes [29,30], the impact of GTR or STR on NKA and NK cell subsets redistribution was also investigated.

## 2. Materials and Methods

### 2.1. Patients

This prospective study recruited 20 patients with histologically confirmed primary or recurrent glioblastoma treated in our institution between January 2020 and May 2021 and enrolled 6 healthy volunteers from our healthy center. The cancer types were determined according to the 2016 World Health Organization Classification of Tumors of the Central Nervous System [31]. Medical records of all participants were reviewed retrospectively, including tumor type, gender, age, and laboratory findings. The inclusion criteria for patients with glioblastoma in this study were (1) aged 20 or older; (2) patients with pathologically confirmed newly diagnosed or recurrent glioblastoma; and (3) receiving surgical resection of tumor. Patients with concomitant autoimmune disease, infectious, inflammatory process and/or with other second or occult tumors were excluded from this study. For healthy subjects, the inclusion criteria were (1) aged 20 years or older at the time of obtaining the informed consent; (2) medically healthy with no significant abnormal screening results clinically, such as vital signs, physical examination, electrocardiograms, and laboratory data; (3) no glioblastoma or other occult tumors; (4) no medical history of tumors; and (5) no previous or concurrent immune disease, infectious process or inflammatory state. Any subjects who did not fulfill the inclusion criteria were excluded.

### 2.2. Ethical Considerations and Surgical Resection of Glioblastoma

This study protocol was approved by the Institutional Review Board of Chang Gung Memorial Hospital (CGMH), Linkou, Taiwan (Number: 201900979B0) and conducted in accordance with the Helsinki Declaration. All included patients and healthy subjects provided signed informed consent to participate. The treatment decision for each patient was evaluated by a multidisciplinary team, including neurosurgeons, radiation oncologists, medical oncologists, neuroradiologists, and neuropathologists. Treatment decisions for each patient were evaluated by a multidisciplinary team including neuropathologists, neurooncologist, neurosurgeons, radiation oncologists, and medical oncologists. All patients underwent cranial surgical resection of glioblastoma. Surgical resection was performed to maximally remove the tumor mass and preserve as much functionally intact brain tissues as possible within the tumor boundaries. All patients received intravenous dexamethasone (Standard Chem & Pharm Co., Ltd., Tainan City, Taiwan) perioperatively (5 mg, q6h). The extent of resection was classified as GTR and STR based percentage of evaluable surgical removal, where GTR was defined as large than 95% resection, while STR was defined as 90–95% resection rate [32]. Peripheral blood samples were collected from glioblastoma patients preoperatively and on postoperative day (POD) 3 and 30.

### 2.3. Blood Sampling and Processing

Venous blood of patients and healthy controls was drawn into BD Vacutainer^®^ Heparin Tubes coated with sodium heparin (BD Biosciences, Becton Dickinson, Franklin Lakes, NJ, USA). One milliliter of whole blood was transferred into NK VUE tube (NKMAX, Seongnam-si, Korea), and then the tube contents were gently mixed. After 20–24 h of incubation at 37 °C, the plasma was collected by centrifugation at 1200× *g* for 4 min and stored at −20 °C. The remaining whole blood was used to isolate peripheral blood mononuclear cells (PBMCs) using Ficoll-Pague^®^ (Cytiva, Marlborough, MA, USA) density gradient centrifugation. The isolated PBMCs were cryopreserved for later analysis.

### 2.4. Determination of NKA and Absolute NK Cell Counts

NKA was determined by measuring the secreted IFN-γ released by NK cells using the NK VUE Kit (NKMAX, Seongnam-si, Korea) according to the manufacturer’s instructions. Briefly, cryopreserved plasma samples were thawed and centrifuged at 11,500× *g* for 1 min at room temperature. Then, the supernatants were transferred into the diluent-loaded ELISA wells, and the mixtures were incubated for 1 h at room temperature. After washing away unbound material, IFN-γ was determined by anti-IFN-γ antibody conjugated to horseradish peroxidase (HRP). Subsequently, tetramethyl benzidine solution was aliquoted and incubated for 30 min following a wash to remove the unbound antibody-HRP complex. Finally, absorbance at 450 nm was measured and the amount of NK cell-secreted IFN-γ was quantitated. Absolute NK cell counts were determined from the peripheral blood of the patients and calculated using the formula WBC (cells/l) * Lymphocytes (%) * CD3^−^CD56^+^ and/or CD3^−^CD16^+^ cells (%).

### 2.5. Flow Cytometery Analysis

Cryopreserved PBMCs were thawed in a 37 °C water bath, and then transferred to a 15 mL centrifuge tube containing 5 mL of PBS. Then, PBMCs were incubated at 37 °C for 5 min, followed by centrifugation at 300× *g* for 10 min. Next, the supernatants were discarded, and PBMCs were resuspended by adding 1 mL of PBS. After 1 h incubation at 37 °C, PBMCs were centrifuged at 300× *g* for 10 min. Subsequently, PBMCs were resuspended and stained by the following monoclonal antibodies (mAbs) for 1 h: anti-CD3-PerCp-Cy5.5 (Catalog No. 560835), anti-CD4-APC (Catalog No. 555349), anti-CD8-FITC (Catalog No. 555366), anti-CD56-PE (Catalog No. 555516) and anti-CD16-BV421 (Catalog No. 562874). All mAbs were purchased from BD Biosciences (Becton Dickinson, Franklin Lakes, NJ, USA). Finally, stained PBMCs were analyzed using BD Fortessa flow cytometer (BD Biosciences, Becton Dickinson, Franklin Lakes, NJ, USA). Appendix A shows the gating strategy for the analysis of NK cell and T cell phenotypes in single live lymphocytes by multiparametric flow cytometry.

### 2.6. Statistical Analysis

Statistical analyses were performed using SPSS software version 22 (IBM Corp., Armonk, NY, USA). Continuous data are presented as median with interquartile range, and categorical data are presented as frequency and percentage. The comparison between healthy controls and glioblastoma patients was statistically analyzed using a two-tailed Mann–Whitney U test. Intergroup comparisons were assessed with the Kruskal–Wallis test, followed by the two-tailed Wilcoxon matched-pairs signed-rank test. *p* values of less than 0.05 were considered statistically significant.

## 3. Results

### 3.1. Baseline Demographic and Clinical Characteristics of Patients with Glioblastoma

Table 1 shows the baseline demographic and clinical characteristics of glioblastoma patients. The median age of these patients was 61.5 years, ranging from 22 to 73 years (mean age: 58.4 ± 13.8 years). Most tumors were recurrent glioblastoma (75%) and located at frontal lobe (55%), parietal lobe (45%), and temporal lobe (20%). Diabetes (25%) was the most common comorbidity. Among them, 11 patients (55%) received gross total resection (GTR) and nine patients (45%) received subtotal resection (STR).

### 3.2. Impaired NKA Recovered 30 Days after Surgical Resection of Glioblastoma

In order to understand whether NKA is impaired in glioblastoma patients, we further recruited healthy subjects and compared the NKA status between glioblastoma patients and healthy control. As shown in Appendix A, glioblastoma patients had a significantly lower NKA than healthy subjects (21.8 pg/mL vs. 874.0 pg/mL, *p* < 0.001), suggesting that NKA is severely impaired in glioblastoma patients.

Next, we analyzed whether surgical resection of glioblastoma affects NKA status. Table 2 shows the NKA, NK count, and distribution of NK-cell and T-cell subsets in glioblastoma patients before surgical resection (baseline) and 3 days (POD3) and 30 days (POD30) after surgery. There were no statistically significant differences in the absolute NK counts and distribution of NK-cell and T-cell subsets between baseline, POD3, and POD30 (*p* > 0.05). However, NKA significantly differs between the baseline, POD3, and POD30 (21.8 pg/mL vs. 7.0 pg/mL vs. 107.6 pg/mL, *p* = 0.001). Figure 1A further shows the statistical analysis between baseline, POD3, and POD30. NKA was significantly decreased three days after surgical resection of glioblastoma (baseline vs. POD3, *p* = 0.002), but was significantly increased 30 days after surgery (baseline vs. POD30, *p* = 0.002). Notably, NKA at POD30 increased nearly five-fold compared with baseline (21.8 pg/mL vs. 107.6 pg/mL, *p* = 0.004), suggesting that surgical resection recovered the impaired NKA in glioblastoma patients. To clarify whether increased NKA after surgery was due to the increase of NK cell number, absolute NK cell counts from peripheral blood were analyzed. As shown in Figure 1B, there were no significant differences in absolute NK cell counts between baseline, POD3, and POD30 (*p* > 0.05).

### 3.3. Redistribution of NK Cell Subsets but Not T Cell Subsets after Cranial Surgery

Given the pleiotropic roles of different NK subsets on tumor immunity [33], we next examined whether surgical resection of glioblastoma affects the distribution of NK cell subsets as well as T cell subset. As shown in Figure 2A, the CD56^bright^CD16^−^ NK subset was significantly decreased on POD3 (median: 1.1% vs. 0.5%, *p* = 0.022, compared with baseline). Moreover, the CD56^bright^CD16^−^ NK subset was instead increased on POD30 (median: 0.5% vs. 0.7%, compared with POD3), although it did not reach statistical significance (*p* > 0.05). Conversely, CD56^dim^CD16^+^ NK subset was significantly increased on POD3 (median: 85.3% vs. 87.5%, *p* = 0.04, compared with baseline; Figure 2B), but further returned to baseline levels on POD30 (median: 85.3% vs. 84.7%, *p* > 0.05). On the other hand, the CD4^+^CD8^−^, CD4^−^CD8^+^, and CD56^+^ T cell populations did not differ significantly between baseline, POD3, and POD30, suggesting that T cell subsets did not redistribute after surgery (All *p* > 0.05, Appendix A).

### 3.4. NKA Is Significantly Increased on POD30 Compared with Baseline in Patients Receiving Gross Total Resection

Extent surgical resection is known to be independently associated with survival outcomes of patients with glioblastoma [34]. Therefore, we next investigated the impact of GTR or STR on NKA, absolute NK cell counts, and distribution of NK and T cell subsets. Table 3 shows the subgroup analysis of extent surgical resection using GTR and STR. Regardless of GTR or STR subgroup, there were no significant differences in the absolute NK cell count and the distribution of NK- and T-cell subsets between baseline, POD3, and POD30 (all *p* > 0.05). However, NKA significantly differed between the baseline, POD3, and POD30 in glioblastoma patients who underwent GTR (7.7 vs. 5.0 vs. 153.5 pg/mL, *p* = 0.001). No significant differences were observed in NKA before and after STR (47.3 vs. 11.4 vs. 51.2 pg/mL, *p* = 0.316). Figure 3 further shows the statistical analysis between baseline, POD3 and POD30. There was no significant difference in NKA between patients before and 3 days after GTR (median: 7.7 vs. 5.0 pg/mL, *p* = 0.155, compared baseline with POD3). Notably, patients receiving GTR resulted in a substantial increase in NKA on POD30 (median: 7.7 vs. 153.5 pg/mL, *p* = 0.015, compared baseline with POD30). In contrast, NKA in patients who received STR was significantly decreased on POD3 (median: 47.3 vs. 11.4, *p* = 0.008, compared baseline with POD3), but returned to baseline on POD30 (median: 47.3 vs. 51.2 pg/mL, *p* = 0.893), suggesting that GTR rather than STR can recover the impaired NKA in patients with glioblastoma.

## 4. Discussion

In this prospective study, we assessed NKA, NK cell counts, and the distribution of NK- and T-cell subsets in glioblastoma patients before and after surgery. Our results showed that glioblastoma patients have extremely low NKA compared with healthy subjects, indicating NK cell dysfunction in glioblastoma patients. Furthermore, surgical resection of glioblastoma not only redistributed NK cell subsets, but also greatly recovered the impaired NKA in glioblastoma patients 30 days after surgery. Stratified analysis further showed that the recovery of impaired NKA was attributable to GTR rather than STR. Therefore, the results of this study suggest that glioblastoma may have a negative impact on NK cell immunity, and that GTR is of great benefit in the recovery of impaired NKA in glioblastoma patients (Figure 4).

Results of the present study found that NKA was further suppressed three days after cranial surgical resection of glioblastoma but recovered and further dramatically increased 30 days after surgery. Early postoperative NKA reduction is thought to be primarily attributable to the physiological response to surgical stress and a cascade of inflammatory responses, such as compensatory anti-inflammatory response [20,35]. Moreover, NK cell suppression after tumor surgery is considered to be a major driver of cancer metastasis and recurrence and is associated with poor survival outcomes [36]. In this study, early NK dysfunction after surgical resection of glioblastoma is also consistent with other studies of tumor resection. A study by Angka et al. [24] showed that NKA was dramatically reduced by 83.1% on POD1 in patients with colorectal cancer and gradually recovered after surgery. In a study of 24 pancreatic cancer patients, NK cell cytotoxicity was found to be significantly downregulated following pancreaticoduodenectomy on POD7 and return to baseline on POD30 [37]. Similar findings were also observed in Velasquez’s study showing that NK function is significantly impaired after surgery for malignant bone tumors without significant changes in NK cell numbers [38]. It is worth noting that NKA in this study was significantly increased on POD30, even higher than baseline values, suggesting that a significant tumor burden had been resected, which may contribute to the gradual recovery of impaired NKA due to the suppression from glioblastoma. In addition, absolute NK cell numbers were not significantly different after surgery. Moreover, the recovery of impaired NKA was not due to an increase in NK cell numbers after surgery. Instead, it may be associated with changes in the redistribution of NK subsets or the quality of NK cells.

In this study, flow cytometric analysis was conducted to investigate whether impaired NKA stemmed from downregulation of CD56^bright^CD16^−^ NK cells, which is a subset of NK cells expressing the largest amount of IFN-γ. As expected, CD56^bright^CD16^−^ NK cells were significantly decreased on POD3, whereas CD56^dim^CD16^+^ NK cells were significantly increased. Although a previous study reported that CD56^dim^ NK cells also expressed IFN-γ within 4 h after triggering NK cell activation, CD56^bright^ NK cells produced the major IFN-γ after 16 h of stimulation [39]. Thus, we speculate that the major IFN-γ production in this study is from CD56^bright^ rather than CD56^dim^ NK cells. In other words, cranial surgery led to a redistribution of NK cell subsets from CD56^bright^CD16^−^ to CD56^dim^CD16^+^ within three days, which in turn lead to NKA downregulation. In addition to cell subset redistribution as a possible cause, the quality of NK cells may also have an impact on NKA. This is because that NKA dramatically increased nearly five-fold at POD30 compared with baseline, but the numbers of CD56^bright^CD16^−^ NK cells did not even return to baseline levels. Despite the current findings, more studies are needed to directly validate whether impaired NKA is due to redistribution and quality of NK cell subsets. On the other hand, several studies have demonstrated that T cells also express IFN- after specific stimulation [40,41,42]. Therefore, we further investigated whether surgical resection of glioblastoma could redistribute T cell subsets. Our data indicated that T cell subsets (CD4^+^CD8^−^, CD4^−^CD8^+^, CD56^+^ T cell) were not affected after cranial surgery, i.e., T cells were not associated with NKA downregulation as a result of surgery.

In terms of potential therapeutics, steroids are often used in combination with cranial surgery, radiation therapy, and palliative care to reduce treatment-related toxicity [43]. Moreover, the use of steroid is known to suppress NK cell functions [44]. An immunophenotyping study by Chitadze et al. found that CD56^bright^ NK cells were significantly downregulated in glioblastoma patients treated with steroids, whereas steroid had no apparent impact on CD56^dim^ NK cells [45]. Vitale further indicated that the surface density of various activating NK receptors declined during methylprednisolone treatment, which is recognized as NK cell dysfunction, and then returned to normal levels shortly after steroid discontinuation or low-dose use [46]. Therefore, in this study, the decline of NKA and CD56^bright^ NK cells on POD3 after surgical removal glioblastoma may not be or slightly due to the steroid use. Nevertheless, in this study, the steroid was stopped about five days after the operation. This suggests that significant increase NKA and recovery of CD56^bright^ NK cells on POD30 may be mainly due to the evacuation of tumor cells, which eliminated the immune inhibition effect, and partly due to cessation of steroid usage.

In a recent comprehensive meta-analysis by Tang et al., GTR is superior to STR in terms of recurrence, survival rates, and functional outcomes in glioma patients [30]. It is generally believed that residual brain tumor cells closed to the resection border may remain alive and eventually reproliferate, leading to rapid recurrence. However, NK cells also play a key role in preventing tumor progression and metastasis through their direct cytotoxic activity and secreted cytokines [47]. Recent studies have shown that low NKA is significantly associated with a higher risk of various cancers, such as hepatocellular carcinoma [36], colorectal cancer [48], head and neck squamous cell carcinoma [49], lung cancer [50], and pancreatic cancer [51]. Jun’s study showed that NKA is progressively impaired during tumor development, and its dysfunction is associated with recurrence and survival outcomes [52]. In this glioblastoma clinical study, we further found that glioblastoma was associated with lower NKA, and tumor resection facilitated NKA recovery, which was mainly attributable to GTR rather than STR. Importantly, during the recovery period after surgery, NKA rises, even above preoperative baseline levels, particularly in patients who received GTR. These results indicate that glioblastoma definitely had a negative impact on the immune system and that the goal of GTR should be achieved when it is possible in order to attain better immune rejuvenation and patient outcomes.

This study has several limitations, including the limited number of cases, which limits further identification of potential confounders associated with impaired NKA in glioblastoma and postoperative NKA recovery. The study was also conducted in a single institution and results may not be generalized to other populations. Some data were reviewed retrospectively from the medical records of prospectively included patients, which may preclude inferences of causality and may also limit long-term follow-up. Longer follow-up of NKA is necessary after completion of concurrent chemoradiation therapy and following treatment. In addition, although we recruited healthy volunteers to examine the NKA levels, incomplete demographic data of healthy subjects on variables may be considered a limitation of this study. Future prospective studies with a large sample size and healthy control are needed to further validate the finding of this study and improve the limitations associated with this study.

## 5. Conclusions

Glioblastoma progression has a great negative impact on the distribution of NK cell subtypes and their activity. NKA is significantly impaired in glioblastoma patients compared with healthy controls. During the recovery period after surgery, GTR rather than STR greatly restored the impaired NKA levels, even several folds higher than preoperative baseline levels. The unsatisfactory effect of STR may be due to continued inhibition of the activity of NK cells by residual tumor closed to the resection border. Therefore, the goal of performing GTR for glioblastoma should be achieved when possible since it appears to increase NK cell immunity. Further investigations are warranted to verify the role and function of these recovered NK cells after GTR in glioblastoma patients and to explore potential confounding factors affecting impaired NKA and GTR-dependent NKA recovery.

## Figures and Tables

**Figure 1 brainsci-12-01144-f001:**
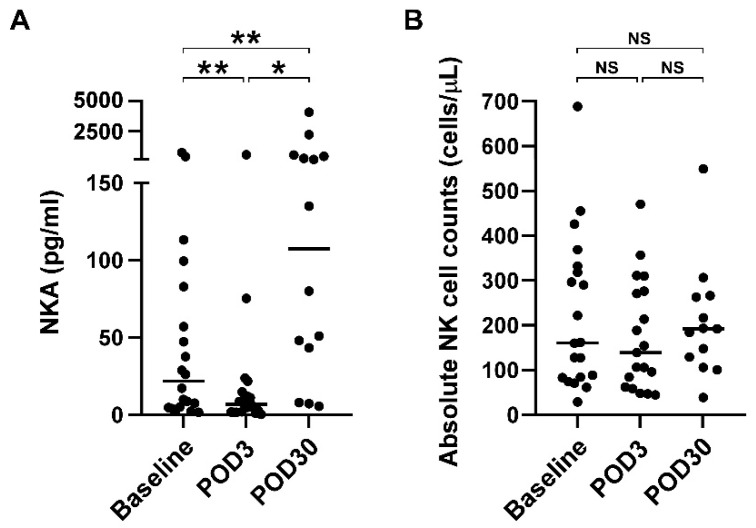
Impaired NKA recovered after surgical resection of glioblastoma on POD30. (**A**) NKA was measured before (baseline) and after surgical resection of glioblastoma on POD3 and POD30. NKA was determined by measuring the NK-released IFN-γ using the NK VUE kit. Data were presented as scatter plot, and differences between groups were statistically analyzed using the two-tailed Wilcoxon matched-pairs signed-rank test. Differences were found to be statistically significant at * *p* < 0.05 and ** *p* < 0.01. Solid line indicates the median. (**B**) Absolute NK cell counts were determined from patients before and after surgical resection of glioblastoma. Data were presented as scatter plot and median, and differences between groups were statistically analyzed using the two-tailed Wilcoxon matched-pairs signed-rank test. NS denotes no statistically significant difference. Abbreviation: NKA, natural killer cell activity; POD3, postoperative day 3; POD30, postoperative day 30.

**Figure 2 brainsci-12-01144-f002:**
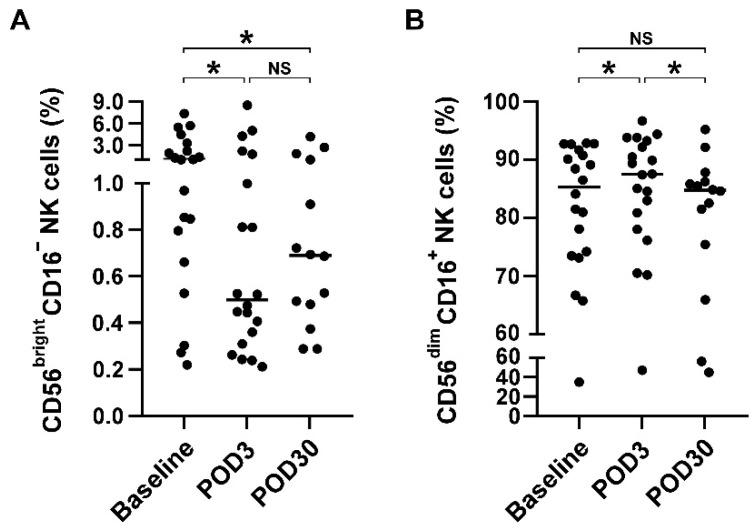
NK cell subsets were redistributed after cranial surgery. (**A**) Distribution of CD56^bright^CD16^−^ NK subsets before and after surgery. Surface expression of CD56 and CD16 were classified by flow cytometric analysis. (**B**) Distribution of CD56^dim^CD16^+^ NK subsets before and after surgery. Data were presented as scatter plot and median, and differences between groups were statistically analyzed using two-tailed Wilcoxon matched-pairs signed-rank test and. Differences were found to be statistically significant at * *p* < 0.05. NS indicates no statistically significant difference. Abbreviation: NK, natural killer; POD3, postoperative day 3; POD30, postoperative day 30.

**Figure 3 brainsci-12-01144-f003:**
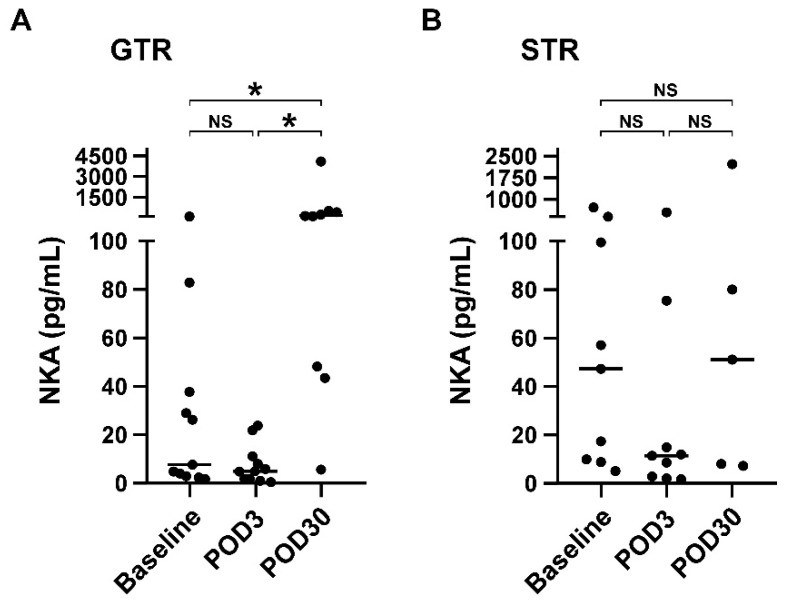
GTR rather than STR significantly recovered the impaired NKA at POD30. NKA of glioblastoma patients before (baseline) and after receiving GTR (**A**) or STR (**B**) on POD3 and POD30. Data were presented as scatter plot with median (solid line), and differences between groups were analyzed using two-tailed Mann-Whitney U test. * *p* < 0.05 was considered statistically significance between groups, while NS denotes no statistically significant difference. Abbreviation: NKA, NK cell activity; GTR, gross total resection; STR, subtotal resection; POD3, postoperative day 3; POD30, postoperative day 30.

**Figure 4 brainsci-12-01144-f004:**
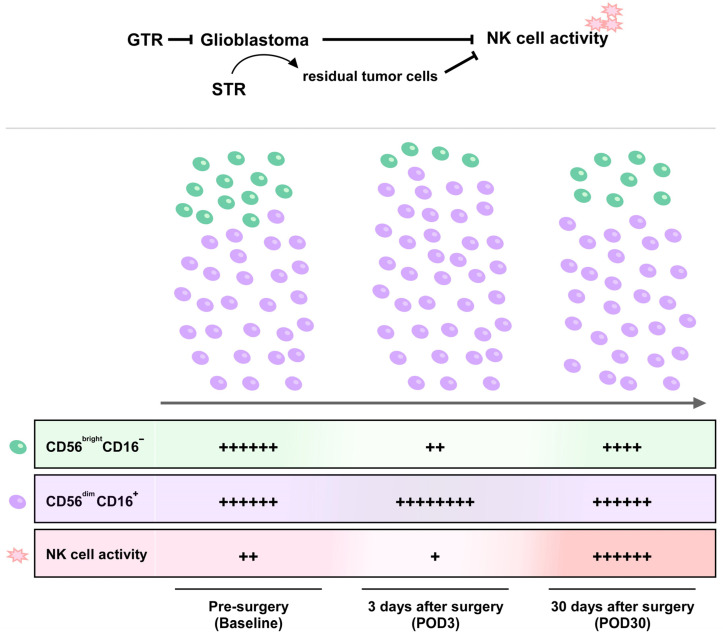
Schematic illustration of GTR in the recovery of the glioblastoma-suppressed natural killer cell activity with further enhancement.

**Table 1 brainsci-12-01144-t001:** Baseline demographic and clinical characteristics of glioblastoma patients (N = 20).

Variable		Glioblastoma Patients
Age, years, median (IQR)		61.5 (52.3–68.8)
Sex		
	Male	9 (45.0%)
	Female	11 (55.0%)
Glioblastoma		
	Primary	5 (25.0%)
	Recurrent	15 (75.0%)
Extent of resection		
	GTR	11 (55.0%)
	STR	9 (45.0%)
Tumor location		
	Frontal lobe	11 (55.0%)
	Parietal lobe	6 (30.0%)
	Temporal lobe	4 (20.0%)
	Insular	1 (5.0%)
	Cerebellum	1 (5.0%)
Comorbidity		
	Diabetes	5 (25.0%)
	Dyslipidemia	3 (15.0%)
	Hypertension	3 (15.0%)
	Asthma	1 (5.0%)
	Polyovarian syndrome	1 (5.0%)
	CSDH	1 (5.0%)
	Gout	1 (5.0%)
	HBV Carrier	1 (5.0%)
	Hepatitis C	1 (5.0%)
	Thyroid goiters	1 (5.0%)
Time points of blood samples		
	Baseline	20 (100%)
	POD3	20 (100%)
	POD30	14 (70.0%)

Continuous data are presented as median with interquartile range, and categorical data are presented as frequency and percentage. Abbreviations: GTR, gross total resection; CSDH, chronic subdural hematomas; HBV, hepatitis B virus; STR, subtotal resection; POD, postoperative days.

**Table 2 brainsci-12-01144-t002:** NKA status and distribution of NK cell and T cell subsets in patients with glioblastoma before and after surgical resection.

		Baseline	POD3	POD30	*p*-Value ^†^
NKA (pg/mL)		21.8 (4.9, 76.5)	7.0 (1.9, 14.2)	107.6 (34.7, 457.9)	0.001
Absolute NK counts (cells/l)		160.99 (83.7, 325.36)	138.85 (62.39, 276.13)	192 (129.06, 262.89)	0.652
NK cell subset (n)					
	CD56^bright^CD16^−^ NK cell (%)	1 (0.7, 3)	0.5 (0.3, 1.6)	0.7 (0.5, 1.2)	0.095
	CD56^dim^CD16^+^ NK cell (%)	1 (0.7, 3)	0.5 (0.3, 1.6)	0.7 (0.5, 1.2)	0.095
T cell subset (n)					
	CD4^+^CD8^−^ T cell (%)	41.3 (32.3, 57)	43.4 (29.8, 57.1)	39.5 (29.2, 68.6)	0.704
	CD4^−^CD8^+^ T cell (%)	5.4 (3.1, 16.6)	7.5 (3.1, 16.8)	5.4 (2.6, 22.3)	0.382
	CD56^+^ T cell (%)	50 (30.7, 55.5)	46.2 (33.2, 59.4)	48.7 (26.7, 56.1)	0.342

Data are presented as median with IQR. ^†^
*p*-value was calculated using the Kruskal-Wallis H-test. Abbreviation: NKA, NK cell activity; NK cells, natural killer cells; POD, postoperative days; STR, subtotal resection; GTR, gross total resection.

**Table 3 brainsci-12-01144-t003:** NKA and NK cells in glioblastoma patients before and after GTR and STR.

	GTR (n = 11)				STR (n = 9)			
	Baseline	POD3	POD30	*p*-Value ^†^	Baseline	POD3	POD30	*p*-Value ^†^
NKA (pg/mL)	7.7 (3, 37.8)	5.0 (1.8, 11.2)	153.5 (45.9, 482.4)	0.001	47.3 (9.4, 250.2)	11.4 (2.5, 45.2)	51.2 (7.7, 1156.8)	0.316
Absolute NK count (cells/mL)	222.2 (83.2, 426.2)	105.7 (62.4, 270.7)	166.1 (117.4, 205.1)	0.619	160.3 (84.2, 296.5)	163.8 (77.3, 310.8)	262.9 (192.0, 266.4)	0.836
NK cell subset								
CD56^bright^CD16^−^ NK cell (%)	1.0 (0.5, 1.9)	0.5 (0.3, 1.8)	0.7 (0.6, 1.4)	0.260	1.4 (0.8, 3.9)	0.5 (0.4, 2.6)	0.4 (0.3, 1.6)	0.160
CD56^dim^CD16^+^ NK cell (%)	88.5 (74.2, 92.7)	89.4 (83, 93.3)	84.9 (82.0, 87.0)	0.478	80.9 (69.9, 90.4)	85.1 (73.2, 91.8)	75.4 (50.6, 89)	0.600
T cell subset								
CD4^+^CD8^−^ T cell (%)	42.8 (30.6, 72.5)	42.7 (22.9, 67.3)	39.8 (34.2, 69.3)	0.984	39.8 (33.9, 54.5)	44.1 (29.9, 52.1)	36.4 (29.2, 52.0)	0.956
CD4^−^CD8^+^ T cell (%)	47.7 (25.6, 54.5)	40.8 (30.1, 51.1)	46.8 (26.7, 55.7)	0.946	52.7 (36.2, 62.3)	50.6 (42, 60.7)	55.2 (43.2, 57.9)	0.998
CD56^+^ T cell (%)	7.4 (3, 23.3)	8.2 (3.2, 19.4)	10.9 (4.6, 23.9)	0.886	3.5 (3.1, 10.3)	3.9 (2.2, 12.4)	2.6 (2.5, 3.5)	0.664

Abbreviation: GTR, gross total resection; STR, subtotal resection; NK, natural killer; NKA, natural killer cell activity; POD3, postoperative day 3; POD30, postoperative day 30. ^†^
*p*-value was calculated using Kruskal-Wallis test.

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
