# Peer review of "Gross Total Resection Promotes Subsequent Recovery and Further Enhancement of Impaired Natural Killer Cell Activity in Glioblastoma Patients"

_brainsci, 2022, doi:10.3390/brainsci12091144_

Round 1

Reviewer 1 Report

Comments and Suggestions for Authors

     1-      Title: Although the title reflects the content, I would suggest rephrasing it in a better way to attract readers. Authors can add something to better reflect the results and show how the inhibition of NK cell activity occurred and due to what.

2-      English: The manuscript could benefit from editing for grammar, missing words, and subject-verb agreement, etc. It is recommended that authors delete irrelevant "general" phrases and sentences, repeated and unneeded words. They should use short sentences. Also, some Introductory sentences are irrelevant or are not needed. There are also typos in the manuscript.

3-      Abbreviations: All abbreviations should be revised and defined at their first use.

4-      Abstract: “P values of less than 0.05 were considered significant.” There is no need to mention this in the abstract.

5-      Abstract: How did the authors recruit the 6 healthy individuals?

6-      Abstract: “NKA was significantly decreased on POD3 (p = 0.002) and recovered on POD30 (p = 0.002).” Did authors measure this in patients’ blood?

7-      Introduction: Authors elaborated extensively on NK cells and subsets, so I suggest moving some parts to the Discussion.

8-      Table 1: Formatting of this Table needs to be revised. Titles of variables can be in bold, and their categories below them not bolded. Authors can refer to this paper to have an idea how to better represent their results: Tables 1, 2, and 3 in https://linkinghub.elsevier.com/retrieve/pii/S1092-9134(21)00024-1. Same applies to Table 2.

9-      Methods: Details on the surgeries done on the patients (total vs. subtotal resection) should be added.

10-  Methods: authors did not include the inclusion and exclusion criteria of the patients. For example, it is not clear whether those patients or healthy individuals had other cancers or had immunodeficiencies which might have affected the results.

11-  References: Many old references used need to be updated.

12-  Table 1: Medical history information of patients should be included. Also, having simple diseases as diabetes would affect the immune system of patients and hence might bias the results.

13-  Table 1: Patient age is confusing. What is 34 and what is 61? Which one is the median and which one is the mean? Also, which age range is the correct one? 28-48 or 56-63?

14-  Results: Authors should add the baseline characteristics of healthy individuals including their medical history, surgical history, and whether they have any immunodeficiencies. In case those individuals have any medical problem, they cannot serve as truly healthy controls to control the results to.

15-  Results: In section “Baseline NKA is decreased in glioblastoma patients”, it is not clear why authors chose to include the median and not the mean. If that is for the low patient number (sample size), including median and CI shows a wide range of values. Did authors have outliers that might have affected the results?

16-  Results: The p-values are missing from Table 2. Authors need to add a column of p-values next to All patients, then another column next to GTR patients, STR patients, and healthy controls. A different table can be also included to compare GTR to STR patients, and each of GTR and STR patients to healthy controls, including p-values. In this case, t-test would be used or ANOVA (when comparing baseline to POD3 and POD30 for each group of patients).

17-  Results: In figures 1 and 2, IFN-gamma and absolute NK cell counts were compared at POD3 and 30 with respect to baseline; however, patients were included as a whole and not stratified into GTR and STR.

18-  Results: In figure 3, why didn’t the authors compare POD3 as well in B and C images?

19-  Results: did the authors adjust their results by eliminating potential confounding factors such as other malignancies that patients have or diseases they have?

20-  Figures: All the figure legends can be revised as to be more informative of the images presented. Also, statistical tests used and meaning of asterix need to be added. Abbreviations used withing Tables and Figures should be defined as well in the legends at the end.

21-  Discussion: Authors should focus more on the main findings and avoid repeating results presentation in the discussion. Authors could also correlate their findings with what has been published in literature. Clinical relevance should be added.

Author Response

Response to Reviewer 1

  1. Title: Although the title reflects the content, I would suggest rephrasing it in a better way to attract readers. Authors can add something to better reflect the results and show how the inhibition of NK cell activity occurred and due to what.

Response: Thanks for the reviewer’s valuable comment. In order to more reflect the results of this study, we amended the title as "Gross total resection restores promotes subsequent recovery and further enhancement of impaired natural killer cell activity in glioblastoma patients".

  1. English: The manuscript could benefit from editing for grammar, missing words, and subject-verb agreement, etc. It is recommended that authors delete irrelevant "general" phrases and sentences, repeated and unneeded words. They should use short sentences. Also, some Introductory sentences are irrelevant or are not needed. There are also typos in the manuscript.

Response: In the revised manuscript, we have carefully corrected these mistakes and delete irrelevant sentences and repeated/unneeded words. In order to further improve the quality of the manuscript, the text of revised manuscript has been edited by a professional medical editor whose mother tongue is English.

  1. Abbreviations: All abbreviations should be revised and defined at their first use.

Response: Thanks for the reviewer’s comment. In the revised manuscript, we have checked/corrected/defined the abbreviations used in this study.

  1. Abstract: “P values of less than 0.05 were considered significant.” There is no need to mention this in the abstract.

Response: We fully agree with this comment. We have deleted this unneeded sentence in the Abstract section of the revised manuscript.

  1. Abstract: How did the authors recruit the 6 healthy individuals?

Response: In this study, all healthy volunteers were recruited from the health center with signed informed consent. In addition, these healthy volunteers have been confirmed by doctors to be free of glioblastoma or occult tumors using brain CT scans and tumor marker screening. The above information and detailed inclusion criteria for healthy subjects (question 10) were added in the Materials and Methods section of the revised manuscript (Paragraph 1 in the Materials and Methods section).

  1. Abstract: “NKA was significantly decreased on POD3 (p = 0.002) and recovered on POD30 (p = 0.002).” Did authors measure this in patients’ blood?

Response: Yes, the nature killer cell activity (NKA) was measured from peripheral blood samples. After collecting the patients’ blood samples at baseline, POD3 and POD30, NK-secreted interferon-γ was determined as NKA using the NK Vue kit. Therefore, we revised this sentences in the Abstract section of the revised manuscript as follows: Peripheral blood samples were collected preoperatively and on postoperative days 3 (POD3) and 30 (POD30), and NKA was measured using the NK VUE kit (revised Abstract section).

  1. Introduction: Authors elaborated extensively on NK cells and subsets, so I suggest moving some parts to the Discussion.

Response: We agree that the description of NK cells and their subsets is too extensive to permit easy summary, and therefore, we rearranged and shortened this part to be more concise and in line with the core of this study. In addition, the paragraphs in the Discussion section were further restructured with relevant supporting references to emphasize clinical relevance.

  1. Table 1: Formatting of this Table needs to be revised. Titles of variables can be in bold, and their categories below them not bolded. Authors can refer to this paper to have an idea how to better represent their results: Tables 1, 2, and 3 in https://linkinghub.elsevier.com/retrieve/pii/S1092-9134(21)00024-1. Same applies to Table 2.

Response: Thanks to the reviewer for the valuable suggestion and example. The format and layout of tables has been revised to better reflect our results.

  1. Methods: Details on the surgeries done on the patients (total vs. subtotal resection) should be added.

Response: Thanks to the reviewer for point out this missing part. The description of treatment decision and extent of resection (GTR and STR) were added in the section 2.2 ethical considerations and surgical resection of glioblastoma, as follows:

The treatment decision for each patient were evaluated by a multidisciplinary team, including neurosurgeons, radiation oncologists, medical oncologists, neuroradiologists, and neuropathologists. Treatment decisions for each patient were evaluated by a multidisciplinary team including neuropathologists, neurooncologist, neurosurgeons, radiation oncologists, and medical oncologists. All patients underwent cranial surgical resection of glioblastoma. Surgical resection was performed to maximally remove the tumor mass and preserve as much functionally intact brain tissues as possible within the tumor boundaries. All patients received intravenous dexamethasone perioperatively (5 mg, q6h). The extent of resection was classified as GTR and STR based percentage of evaluable surgical removal, where GTR was defined as large than 95% resection, while STR was defined as 90-95% resection rate (section 2.2 in the Materials and Methods).

  1. Methods: authors did not include the inclusion and exclusion criteria of the patients. For example, it is not clear whether those patients or healthy individuals had other cancers or had immunodeficiencies which might have affected the results.

Response: Thanks to the reviewer for pointing out the missing part. The following inclusion/exclusion criteria for healthy subjects and patients with glioblastoma were added in the Method section of the revised manuscript (section 2.1).

The inclusion criteria of healthy subjects were 1) aged 20 years or older at the time of obtaining the informed consent; 2) medically healthy with no significant abnormal screening results clinically, such as vital signs, physical examination, electrocardiograms, and laboratory data; 3) no glioblastoma or other occult tumors; 4) no medical history of tumors; and 5) no previous or concurrent immune disease, infectious process or inflammatory state. Any subjects who did not fulfill the inclusion criteria were excluded. The inclusion criteria for patients with glioblastoma in this study were 1) aged 20 or older; 2) patients with pathologically confirmed newly-diagnosed or recurrent glioblastoma; and 3) receiving surgical resection of tumor. Patients with concomitant autoimmune disease, infectious, inflammatory process and/or with other second or occult tumors were excluded from this study.

  1. References: Many old references used need to be updated.

Response: In the revised manuscript, references have been updated to recent years.

  1. Table 1: Medical history information of patients should be included. Also, having simple diseases as diabetes would affect the immune system of patients and hence might bias the results.

Response: Thanks for the reviewer’s comment. We added the medical history to the Table 1 of the revised manuscript. Of these patients, diabetes mellitus (25%) was the most common comorbidity, followed by dyslipidemia (15%) and hypertension (15%).

  1. Table 1: Patient age is confusing. What is 34 and what is 61? Which one is the median and which one is the mean? Also, which age range is the correct one? 28-48 or 56-63?

Response: We apologize for the inadvertent confusion caused by the layout of the table. The table was corrected and re-layout to make it clearer and more concise. The median age of the glioblastoma patients is 61.5 years, with an interquartile range (IQR) of 52.3-68.8. For mean age calculation, the mean age of these patients is 58.4 ± 13.8 years, ranging from 22 to 73 years. The information was described in the Result section of the revised manuscript (paragraph 1 in the section 3.1)

  1. Results: Authors should add the baseline characteristics of healthy individuals including their medical history, surgical history, and whether they have any immunodeficiencies. In case those individuals have any medical problem, they cannot serve as truly healthy controls to control the results to.

Response: This is a very good comment. In this study, our initial purpose of recruiting healthy subjects was simply to understand whether NKA is impaired in glioblastoma patients, and whether the impaired NKA in patients with glioblastoma recovered after GTR or STR surgery. Therefore, we did not examine or record the detailed clinical data of healthy volunteers. Nonetheless, healthy volunteers had to meet the following inclusion criteria: 1) aged 20 years or older at the time of obtaining the informed consent, 2) medically healthy with no significant abnormal screening results clinically, such as vital signs, physical examination, electrocardiograms, and laboratory data; 3) no glioblastoma or other occult tumors, 4) no medical history of tumors; and 5) no previous or concurrent immune disease, infectious process or inflammatory state. The above inclusion criteria of healthy subjects were added to the Methods section of revised manuscript (section 2.1).

  1. Results: In section “Baseline NKA is decreased in glioblastoma patients”, it is not clear why authors chose to include the median and not the mean. If that is for the low patient number (sample size), including median and CI shows a wide range of values. Did authors have outliers that might have affected the results?

Response: We all know that the sampling distribution of the sample means approaches a normal distribution as the sample size gets larger. Therefore, for cases where the sample size is not very large, the median value becomes more important because it gives us an idea of where the center value is located in a dataset. In other words, the median tends to be more useful to calculate than the mean because median can reduce the effect of skewed distribution and/or having outliers. In this study, two patients had outliers in NKA values that greatly increased the mean values. If the two outliers are not included in the calculation, the mean of NKA in POD3 would drop from 38.5 to 7.7 pg/mL, closer to the central value in the original dataset (median value for all patients is 7.0 pg/mL). Therefore, in this study, median is a more appropriate measure of central tendency.

  1. Results: The p-values are missing from Table 2. Authors need to add a column of p-values next to All patients, then another column next to GTR patients, STR patients, and healthy controls. A different table can be also included to compare GTR to STR patients, and each of GTR and STR patients to healthy controls, including p-values. In this case, t-test would be used or ANOVA (when comparing baseline to POD3 and POD30 for each group of patients).

Response: We sincerely thank the reviewer for this valuable comment. In the revised manuscript, we reorganized the tables into Table 1 (presenting the patients’ demographic), Table 2 (presenting NKA changes between baseline, POD3, and POD30), and Table 3 (comparing the GTR and SRT). In addition, Kruskal-Wallis H-test and Wilcoxon signed-rank test were used to compare the differences between groups. The P-value was added in the rightmost column. We greatly appreciate the reviewer's suggestion to make the new tables more concise, clear, and more reflective of statistical significance.

  1. Results: In figures 1 and 2, IFN-gamma and absolute NK cell counts were compared at POD3 and 30 with respect to baseline; however, patients were included as a whole and not stratified into GTR and STR.

Response: We agree that the original formatting may cause inadvertent confusion. Based on above comments, the tables and figures were further re-organized and re-layout to express [1] impaired NKA in glioblastoma patients (Supplementary Figure 1); [2] demographic of glioblastoma patients (Table 1); [3] changes of NKA and redistribution of NK- and T-cell subsets between baseline, POD3 and POD30 (Table 2); [4] recovery of impaired NKA after surgical resection of glioblastoma (Figure 1); [5] redistribution of NK cell subsets before and after surgery (Figure 2); [6] subgroup analysis of GTR and STR (Table 3); and [7] GTR rather than STR significantly recovered the impaired NKT on POD30. We believe that the reorganized tables/figures will help readers better understand the results of this study.

  1. Results: In figure 3, why didn’t the authors compare POD3 as well in B and C images?

Response: Based on the comment, we added the POD3 in the revised Figure 3.

  1. Results: did the authors adjust their results by eliminating potential confounding factors such as other malignancies that patients have or diseases they have?

Response: This is indeed a very interesting question, but unfortunately, due to the research limitations of this study, this statistical analysis cannot be performed now. The initial core design of this study was to provide insights of impaired NK cell activities in patients of glioblastoma and the benefit of GTR in the recovery of NK cell activities. However, the issue of potential confounding factors is indeed another important scientific research that requires future prospective or retrospective multicenter studies and recruitment of more patients. Therefore, we added this as one of the limitations of this study. Nonetheless, present findings provide a rationale and study basis for our two future studies: What are the molecular mechanisms by which GTR recovers the impaired NKA within 30 days? Are there any confounding variables influencing the impaired NKA in glioblastomas and GTR-dependent recovery?

  1. Figures: All the figure legends can be revised as to be more informative of the images presented. Also, statistical tests used and meaning of asterix need to be added. Abbreviations used within Tables and Figures should be defined as well in the legends at the end.

Response: Thanks for the reviewer’s comment. As suggested, we have supplemented the legend with more information about the figures and tables, including more detailed descriptions, meaning of the asterisks (statistical analysis), abbreviations, etc.

  1. Discussion: Authors should focus more on the main findings and avoid repeating results presentation in the discussion. Authors could also correlate their findings with what has been published in literature. Clinical relevance should be added.

Response: Based on the valuable comment, we reorganized the paragraphs in Discussion section and supplemented with relevant supporting references to emphasize clinical relevance, as follows: (1) Paragraph 1 addressed the study purpose and summarized the main finding of this study; (2) Paragraphs 2 and 3 described possible reasons for the observed decline in NKA and redistribution of NK subtypes on POD3, along with similar supporting evidence from other published literature; (3) The effect of steroid use on postoperative NKA was discussed in paragraph 4; (4) Paragraph 5 discussed the intraoperative and postoperative use of steroid, which may be contributed to the NKA inhibition observed at POD3; (5) Paragraph 6 discussed the benefits of GTR for maximal extent of resection of residual tumors and recovery of NKA; and (6) The limitations of this study in paragraph 7. Finally, we would like to express our thankfulness to the anonymous reviewer’s valuable comments in improving the quality of this study.

Reviewer 2 Report

Comments and Suggestions for Authors

Reviewer (Remarks to the Author):

The manuscript entitled “Inhibition of Natural Killer Cell Activity in Glioblastoma Patients” by Huang et al. demonstrate that NKA increased significantly on POD30 in patients. The premise of the work is very interesting, however in its present version, the manuscript requires several significant areas of improvement before consideration for publication.

1) It would be more interesting if authors can provide one separate model Figure showing the results of manuscript.

2) The descriptions of data in the figures also needs to be significantly improved.

3) Although the manuscript shows some interesting correlative data, there are some issues with quality of Figures, Tables. The quality of Figures, and Tables needs to improved.

4) I question if the "results" section needs to be so lengthy and detailed. It is difficult for readers who don't have a strong basic sciences foundation to understand this section.

5) There are some points either discussed haphazardly or overlooked, need to be discussed properly. I wonder if authors would provide a box (containing some bullet points) addressing some major points/ mechanisms/ challenges and/ or answers of some demanding questions of the discussed area.

6) The text needs careful proof reading.

 Reviewer (Remarks to the Author):

The manuscript entitled “Inhibition of Natural Killer Cell Activity in Glioblastoma Patients” by Huang et al. demonstrate that NKA increased significantly on POD30 in patients. The premise of the work is very interesting, however in its present version, the manuscript requires several significant areas of improvement before consideration for publication.

1) It would be more interesting if authors can provide one separate model Figure showing the results of manuscript.

2) The descriptions of data in the figures also needs to be significantly improved.

3) Although the manuscript shows some interesting correlative data, there are some issues with quality of Figures, Tables. The quality of Figures, and Tables needs to improved.

4) I question if the "results" section needs to be so lengthy and detailed. It is difficult for readers who don't have a strong basic sciences foundation to understand this section.

5) There are some points either discussed haphazardly or overlooked, need to be discussed properly. I wonder if authors would provide a box (containing some bullet points) addressing some major points/ mechanisms/ challenges and/ or answers of some demanding questions of the discussed area.

6) The text needs careful proof reading. 

Author Response

Reviewer 2

1) It would be more interesting if authors can provide one separate model Figure showing the results of manuscript.

Response: Based on the reviewer’s comment, we added a new Figure 4 to schematically illustrate that GTR can restore glioblastoma-suppressed natural killer cell activity with further enhancement, while residual tumor cells after STR may still continue to suppress natural killer activity.

2) The descriptions of data in the figures also needs to be significantly improved.

Response: Thanks for the reviewer’s comment. In the revised manuscript, we have supplemented the legend with more information about the figures and tables, including more detailed descriptions, meaning of the asterisks in statistical analysis, abbreviations, etc.

3) Although the manuscript shows some interesting correlative data, there are some issues with quality of Figures, Tables. The quality of Figures, and Tables needs to improved.

Response: In order to make it easier for reviewers and readers to understand this study, the tables and figures were further re-organized and re-layout to express [1] impaired NKA in glioblastoma patients (Supplementary Figure 1); [2] demographic of glioblastoma patients (Table 1); [3] changes of NKA and redistribution of NK- and T-cell subsets between baseline, POD3 and POD30 (Table 2); [4] recovery of impaired NKA after surgical resection of glioblastoma (Figure 1); [5] redistribution of NK cell subsets before and after surgery (Figure 2); [6] subgroup analysis of GTR and STR (Table 3); and [7] GTR rather than STR significantly recovered the impaired NKT on POD30. We believe that the reorganized tables/figures will help readers better understand the results of this study.

4) I question if the "results" section needs to be so lengthy and detailed. It is difficult for readers who don't have a strong basic sciences foundation to understand this section.

Response: We agree with the reviewer’s comment. We therefore re-described in main text why these experiments were performed and the results obtained. Together with the reorganization of the figures and tables, we believe this added information can help readers better understand the logical thinking and implications of the experiments in our study.

5) There are some points either discussed haphazardly or overlooked, need to be discussed properly. I wonder if authors would provide a box (containing some bullet points) addressing some major points/ mechanisms/ challenges and/ or answers of some demanding questions of the discussed area.

Response: We really appreciate the reviewer’s advice. First and foremost, we added the following highlights with three bullet points to better understanding the content of our study: 1) Natural killer cell activity is dramatically impaired in patients with glioblastoma; 2) Surgical resection of glioblastoma promotes redistribution of NK cell subsets and increases NK cell activity 30 days after surgery; and 3) Gross total resection rather than subtotal resection significantly recovers and further increased the impaired NK cell activity in patients with glioblastoma. (Page 1, below the abstract section).

On the other hand, after carefully reading the discussion paragraphs, we reorganized the paragraphs in Discussion section and supplemented with relevant supporting references to emphasize clinical relevance, as follows: Paragraph 1, addressing the study purpose and summarized the main finding of this study; Paragraphs 2 and 3 to describe the possible reasons for the observed decline in NKA and redistribution of NK subtypes on POD3, along with similar supporting evidence from other published literature; Paragraph 4 to discuss the effect of steroid use on postoperative NKA; Paragraph 5 discussed the intraoperative and postoperative use of steroid, which may be contributed to the NKA inhibition observed at POD3; Paragraph 6 discussed the benefits of GTR for maximal extent of resection of residual tumors and recovery of NKA; Paragraph 7 described the limitations of this study.

6) The text needs careful proof reading.

Response: Thanks for the reviewer’s comment. We have carefully corrected the mistakes throughout the revised manuscript. In addition, the text of final revised manuscript has been edited by a professional medical editor whose mother tongue is English to further improve the quality of the manuscript. Finally, we would like to thank the anonymous reviewer for your valuable comments to improve the quality of this study.